# Cybersecurity Behavior among Government Employees: The Role of Protection Motivation Theory and Responsibility in Mitigating Cyberattacks

**Noor Suhani Sulaiman [1,*], Muhammad Ashraf Fauzi [1,*] , Suhaidah Hussain [1] and Walton Wider [2]**

[1]    Faculty of Industrial Management, University Malaysia of Pahang (UMP), Gambang 26300, Malaysia
[2]    Faculty of Business and Communications, INTI International University, Nilai 71800, Malaysia
*    Correspondence: suhani.sulaiman@gmail.com (N.S.S.); ashrafauzi@ump.edu.my (M.A.F.)

**Abstract:** This study examines the factors influencing government employees' cybersecurity behavior in Malaysia. The country is considered the most vulnerable in Southeast Asia. Applying the protection motivation theory, this study addresses the gap by investigating how government employees behave toward corresponding cyberrisks and threats. Using partial least-squares structural equation modeling (PLS-SEM), 446 respondents participated and were analyzed. The findings suggest that highly motivated employees with high severity, vulnerability, response efficacy, and self-efficacy exercise cybersecurity. Incorporating the users' perceptions of vulnerability and severity facilitates behavioral change and increases the understanding of cybersecurity behavior's role in addressing cybersecurity threats—particularly the impact of the threat response in predicting the cybersecurity behavior of government employees. The implications include providing robust information security protection to the government information systems.

**Keywords:** cybersecurity behavior; protective motivation theory; government employee; threat awareness; protection habit

## 1. Introduction

The growth of technology has increased the number of internet users. Businesses rely entirely on the infrastructure of computer networks and underlying information systems. Organizations' reliance on information technology and the widespread use of the internet and computers necessitate the ongoing access to vital data. Such data are exposed to external cybercrime threats and the risk of information leakage [1]. The perpetrators of this cybercrime can use personal and organizational expertise, such as intellectual property and consumer data, to participate in illicit operations, such as data manipulation, for financial gain [2]. An organization must establish stringent, proactive cybersecurity procedures to safeguard their data and limit the possibilities of any data breaches [3].

Cybersecurity is a broad field that encompasses both technological and human elements. Most security breaches occur due to poor judgment, user mistakes, or a combination of the two [4]. However, poorly behaved computer users can jeopardize organizations. Numerous privacy breaches have affected millions of individuals worldwide [2]. Network intrusions include the user's lack of awareness, ignorance, negligence, resistance, apathy, and mischievousness [5]. Such reported data breaches emphasize the importance of adequate security awareness training to increase computer users' comprehension and drive positive behavioral changes.

Malaysia's internet users accounted for 87.4% of the population, ranking second in the region, and were identified as the most vulnerable to cyberattacks [6]. Malaysians have become an easy target for cybercrime, fraud, and phishing due to a lack of understanding, an inability to curb, and a failure to recognize an actual threat. Computer users' actions in defending and securing resources create vulnerabilities in the network's infrastructure.

Inadequate and inconsistent user behavior can be costly for businesses. Additionally, research reveals that businesses experience at least one data breach [7–9]. The increasing number of data breaches caused by human mistakes has altered the industry's perspective on information security [10].

To understand the determinants of an employee's cybersecurity behavior, this study examines the protection motivation theory (PMT) domains: threat appraisal and coping appraisal. This study aims to establish a link between user awareness and threat appraisal when a cyberattack is detected and the impact of habit in influencing user coping evaluation.

## 2. Literature Review

In recent years, the science of cyberbehavior has advanced tremendously [11]. Numerous studies have been conducted on cybersecurity behavior, most notably on human behavior, cognition, and emotion [12]. No matter how sophisticated an organization is and how much money they spend on security systems, the human element will always be the weakest link, prone to failure and error [13]. Although previous research has established a link between human characteristics and ineffective security practices, a thorough grasp of the subject remains uncommon and limited [14]. Individuals who are more likely to fall victim to cybercrime can be recognized, which benefits researchers, network security employees, organizations, and the nation.

Cybercriminals have few predetermined aims when they conduct cyberattacks. Unauthorized banking activities and credit card usage are at the top of the list [15]. Other offences include the installation of ransomware, theft of medical records, and intellectual property infringement [16]. As consumers, governments, and business owners rely extensively on the internet and cyberinfrastructure, there are growing concerns about harmful cyberactivity [17]. While user awareness has increased and expanded in lockstep with emerging technology via media and press coverage on cyberthreats, most users are exposed to unknown threats and risks [18]. These threats result from criminals constantly evolving their ideas and tactics for duping potentially vulnerable consumers.

While developing countries fundamentally lag behind their industrialized counterparts in terms of acceptance and internet use, they account for roughly two point five billion mass users compared to the latter's one billion [19]. As a developing country transitioning to developed status, Malaysia has one of the highest internet penetration rates in the region, with the Malaysian population increasing from 76.9% in 2016 to 87.4% [20]. Having a strong security system in place and an advanced understanding among the population on how to deal with cyberattacks is critical for society and the country's economic progress. Thus, understanding cybersecurity is deemed critical in recognizing threats (threat appraisal) and removing threats (coping appraisal). These domains fall under the umbrella of the PMT, which serves as the underpinning theory in this study.

### 2.1. Protection Motivation Theory (PMT)

The theory refers an individual's cognitive capability and processes for mediating a specific behavior in the face of danger [21]. When individuals perceive a threat, they focus on two assessment processes: the threat and their ability to overcome it. This is referred to as threat and coping evaluation.

Threat appraisal is made up of two components, which are perceived severity and perceived vulnerability. In the context of this study, perceived severity is a judgment that one makes when confronted with a threat that could result in severe damage at work or anywhere else with an internet connection [22]. If an employee perceives the seriousness of a potential attack, they may engage and take preventative action by attending training or adapting their security knowledge and practices. Individuals' perceptions of vulnerability reflect their fear of a cyberattack and their knowledge that they lack preventative procedures to thwart such attacks [1]. On the other hand, individuals with prior experience dealing with cyberattacks are less likely to be vulnerable.

A coping appraisal consists of three domains. Perceived barriers vary according to an individual's experience in dealing with cyberattacks and having prior experience would motivate them to take cybersecurity protective and preventative measures. Response efficacy is the degree to which an individual believes that a recommended response will effectively mitigate their threat [23]. They will know who can counter cyberthreats based on their experience. On the other hand, self-efficacy refers to an individual's impression of their ability to withstand a cyberattack by taking adequate precautions and coping with the threat [1].

Previous studies using PMT have shown its potential in predicting people's computer-safe behavior at home and in organizational contexts [24–26]. As a result, PMT has been used in various cybersecurity studies. There have been studies on internet users' awareness [27], undergraduate college students' desktop security behavior [28], college students' protective behavior via personal responsibility [29], online security behavior [30], and employee protection on organization information assets [31].

On the whole, based on the scant available literature on user cybersecurity behavior among government employees, this study presents a framework model comprising two significant areas that directly impact user cybersecurity behavior among Malaysian government employees by incorporating user habits and threat awareness as the antecedents of threat and coping appraisal.

### 2.2. Hypothesis Development and Research Model

#### 2.2.1. Threat Awareness and Perceived Severity

Awareness of information security or comprehension of the threat to information security may result in one or more information security responses [32]. Hughes [33] found that raising employee understanding and enforcing information security regulations can help enhance organizational security attitudes. Additionally, organizations must thoroughly understand information security, including monitoring and evaluating the information security program [34]. Thus, research on information security awareness and communication has demonstrated that consistent communication about information security can benefit the organization [35]. Additionally, organizational insiders with a high level of organizational identification may feel more concerned by the repercussions of cyberattacks on the organization than those with a low level of organizational identification, even if their perceived severity is the same. An individual is aware of feasible countermeasures to such risks and should assess their feasibility in practice [28]. Thus, it is proposed that threat awareness has a positive effect on the user's perception of severity:

**H1.** *Threat awareness has a positive influence on the user's perceived severity.*

#### 2.2.2. Threat Awareness and Perceived Vulnerability

Individual cybersecurity awareness, which may be usefully classified as low, medium, or high, is critical in determining information security risk. Awareness-raising entails familiarity with cyberdangers and the ability to prevent them [36]. Shaw et al. [37] described that users recognize the value of information security and its associated obligations and employ suitable information security management mechanisms to safeguard organizational data and network behavior. Meanwhile, an employee more exposed to his organization's information systems will be more prepared to take preventative measures specified as perceived vulnerability. Numerous analyses have revealed that workers' perceived vulnerability to cyberattacks pushes them to embrace cybersecurity practices [3,38]. Thus, it is proposed that threat awareness will positively influence a user's perceived vulnerability.

**H2.** *Threat awareness has a positive influence on the user's perceived vulnerability.*

### 2.2.3. Protection Habits and Perceived Barrier

Habit is a significant antecedent of individual cybersecurity behavior [1]. Habit refers to a pattern of conduct measured using previous behavior or behavioral frequency. Integrating habits within the coping appraisal domain has been shown to predict individual cybersecurity behavior [39]. Perceived barriers are utilized to determine the inconvenience associated with cybersecurity issues. It is inextricably linked to preventive protection behavior. The perceived barriers to behavior change are critical when analyzing behavioral change [40]. In addition, perceived impediments add to the perceived cost and the complexity of conducting cybersecurity operations individually [25]. Individuals' perceptions of data protection challenges in cybersecurity are based on their prior experiences. As cybersecurity prevention methods were implemented, it was observed that the perceived low barriers were influenced by their prior experience. Thus, the following hypothesis is presented:

**H3.** *Protection habits have a negative influence on the user's perceived barrier*.

### 2.2.4. Protection Habits and Response Self-Efficacy

Within the PMT framework, a deeper understanding of the behaviors and routine of those behaviors, which is protection habits, could assist in delivering new insights and increasing the model's predictive potential [41]. Prior research has examined habit from three viewpoints in light of these arguments: the moderating influence of habit on the relationship between intention and information technology usage, the direct effect of habit on information technology use, and the direct effect of habit on intentions to use information technology [42]. Vance et al. [1] stated that many acts occur automatically and are completed because individuals are accustomed to completing them; frequently repeated behavior is more influenced by situational signals than conscious decision-making. In the context of cybersecurity awareness, the response efficacy refers to the amount workers believe that the proposed remedies will significantly reduce their level of risk [43] and the effectiveness of the recommended behavior in eradicating or averting possible harm [44]. Additionally, the same study discovered a substantial correlation between employee response efficacy and their plans to adopt cybersecurity measures. Hence, the following hypothesis is drawn:

**H4.** *Protection habits have a positive influence on the user's response self-efficacy*.

### 2.2.5. Protection Habits and Security Response Efficacy

Security response efficacy relates to an individual's belief in the effectiveness of protective action in repelling a threat. Individuals skeptical about the efficacy of protective action as a security reaction are less likely to utilize it [41]. An argument in a fear appeal will drive an individual to create efficacy cognitions [45]. The former is referred to as security response efficacy, which relates to how effectively a person believes the reaction is addressing a cyberthreat [46]. Meso et al. [44] stated that it is the belief that the proposed conduct can be successfully implemented. It is an individual's assessment of their capacity to withstand a cybersecurity attack by implementing proper safeguards and coping with the threat [1]. A positive cybersecurity behavioral habit can help prevent and protect individuals from cyber-related incidents.

Meanwhile, harmful cybersecurity behavioral habits can expose individuals to additional cybersecurity threats. However, some studies in information security behavior have recognized the importance of habits [30,47]. Findings suggest habits have been found to predict intentions to follow information system security policies. A person with a high level of reaction efficacy regarding cybersecurity behaviors will strongly believe in his or her ability to avert cyberattacks successfully. This will increase the likelihood of previously practiced cybersecurity activities being repeated with a goal in mind [48]. As a result, we proposed the following hypothesis:

**H5.** *Protection habits have a positive influence on the user's security response efficacy*.

### 2.2.6. Perceived Severity and Cybersecurity Behavior

Perceived severity can be defined as the individual perception of an event's consequence [44]. It is related to trust in the magnitude of the consequences of the circumstances [49]. Previous research indicates that the perceived severity of the penalty has a detrimental effect on one's willingness to abuse information systems [50]. The perceived severity is determined in response to a dangerous security event [30]. The higher one's perceived threat, the more probable it is that online users will be protected [29]. Employee's perceptions of the severity of cyberrisks significantly impact their safety concerns [51]. Subsequently, perceived severity effectively reduces information infrastructure misuse [52]. Furthermore, individual behavior depends on the perceived severity of the impact, which increases a consumer's desire to take action to mitigate the threat, thus reducing the likelihood of threats [53,54]. Hence, we propose the following hypothesis:

**H6.** *Perceived severity has a positive influence on the user's cybersecurity behavior.*

### 2.2.7. Perceived Vulnerability and Cybersecurity Behavior

Employee vulnerability is another dimension of cybersecurity threat assessment. Employees who believe their company's information system is at risk are more likely to take preventive steps. According to certain studies, employees' perceived vulnerability to cyberattacks motivates them to adhere to cybersecurity regulations [38]. Perceived vulnerability is a person's assessment of the possibility of being confronted with threatening situations, such as becoming a victim of cybercrime [3]. For example, in e-mail security, perceived vulnerability to malicious attachments has been connected to computer security behavior [55]. As a result, De Kimpe et al. [56] hypothesized that perceived cybercrime vulnerability would be linked to a protective motive in the same way.

**H7.** *Perceived vulnerability has a positive influence on the user's cybersecurity behavior.*

### 2.2.8. Perceived Barrier and Cybersecurity Behavior

Perceived barriers refer to an employee's perceived annoyance and cost associated with cybersecurity protection activities [22]. It was found that perceived barriers have a detrimental effect on users' behavior in computer security [57]. Perceived barriers can be depicted in terms of aggravation, time constraints and addiction that influence user behavior. Employees consider perceived hurdles a challenge they must overcome, lowering their commitment to comply with cybersecurity regulations and take preventative action [22]. Therefore, the following hypothesis is developed:

**H8.** *Perceived barrier has a negative influence on the user's cybersecurity behavior.*

### 2.2.9. Response Self-Efficacy and Cybersecurity Behavior

Situational supports such as interpersonal assistance, supervisor or colleague support, and appropriate time for practice behaviors would be conducive to individual self-efficacy in information security behaviors [58]. Self-efficacy benefits from using protective information technologies [59]. Individuals' decisions to engage in cybersecurity behavior are influenced by their perception of the probability of good consequences [60]. Surroundings and the environment provide individuals adequate situational support to increase self-efficacy, influencing information security practices [48]. Thus, the following hypothesis is proposed:

**H9.** *Response self-efficacy has a positive influence on the user's cybersecurity behavior.*

### 2.2.10. Security Response Efficacy and Cybersecurity Behavior

Users who benefit from better privacy due a personalization platform will have fewer privacy concerns specific to the system and will be more receptive to using it as a tool

for privacy [61]. Self-efficacy and response efficacy are critical components of frameworks that attempt to explain the process by which cybersecurity activities become habits. They are frequently used to explain the establishment of information security behaviors [62]. Individuals with high self-efficacy to engage in cybersecurity behaviors will be strongly correlated. Consequently, individuals demonstrating a high level of reaction efficacy will firmly believe in their ability to prevent cyberattacks. This will increase the likelihood of previously practiced cybersecurity activities with a predetermined goal [48]. Based on this, the following hypothesis is developed:

**H10.** *Security response efficacy has a positive influence on the user's cybersecurity behavior.*

The following Figure 1 illustrates the study research model.

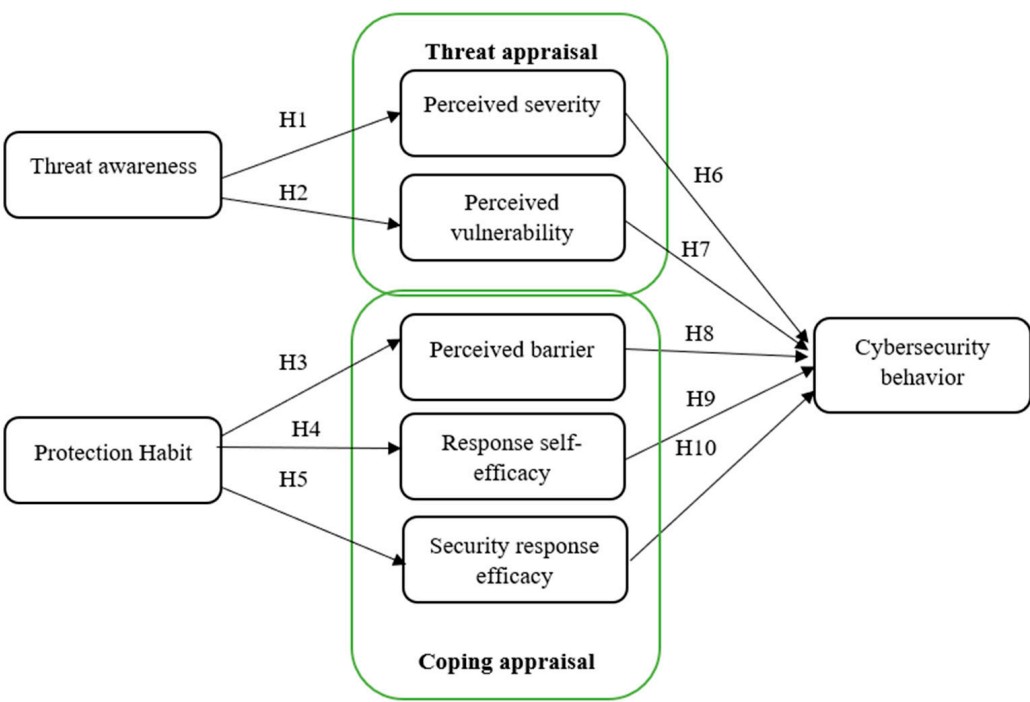

**Figure 1.** Research model.

### 3. Methodology

*3.1. Data Collection*

This study used a quantitative research design through a self-administered questionnaire. The study used an online poll to collect data from a representative sample of government employees in Malaysia who were also internet users. A quota sampling of 20:30:50 was used to collect 446 respondents from government employees of Malaysia. A total of 78 respondents were gathered from top management, 229 respondents were collected from middle management, and 325 were obtained from supporting staff.

GPower 3.1.9.7 software [63] was used in this study to determine the population impact measurement based on the model to be tested. This software can calculate the minimum number of respondents needed for the survey. This study required a minimum sample size of 153 samples. Figure 2 depicts the analysis performed with GPower software to determine the sample size for testing the model. Thus, based on the sample size calculation (Krecjie & Morgan, 1970), 384 samples will be obtained out of the 446 samples to be collected.

$$s = X^2 N P (1 - P)/d^2 (N - 1) + X^2 P (1 - P)$$

where,

$s$—Required sample size.

$X^2$—The table value of chi-square for 1 degree of freedom at the desired confidence level (0.05 = 3.841).

$N$—The population size.

$P$—The population proportion (assumed to be 0.50 since this would provide the maximum sample size.

$D$—The degree of accuracy expressed as proportion (0.05).

$$s = \frac{3.841^2(1,600,000 \times 0.5)(1{-}1,600,000)}{0.05^2(1,600,000 - 1) + 3.841^2 \times 1,600,000(1{-}1,600,000)}$$

**s = 384 samples**

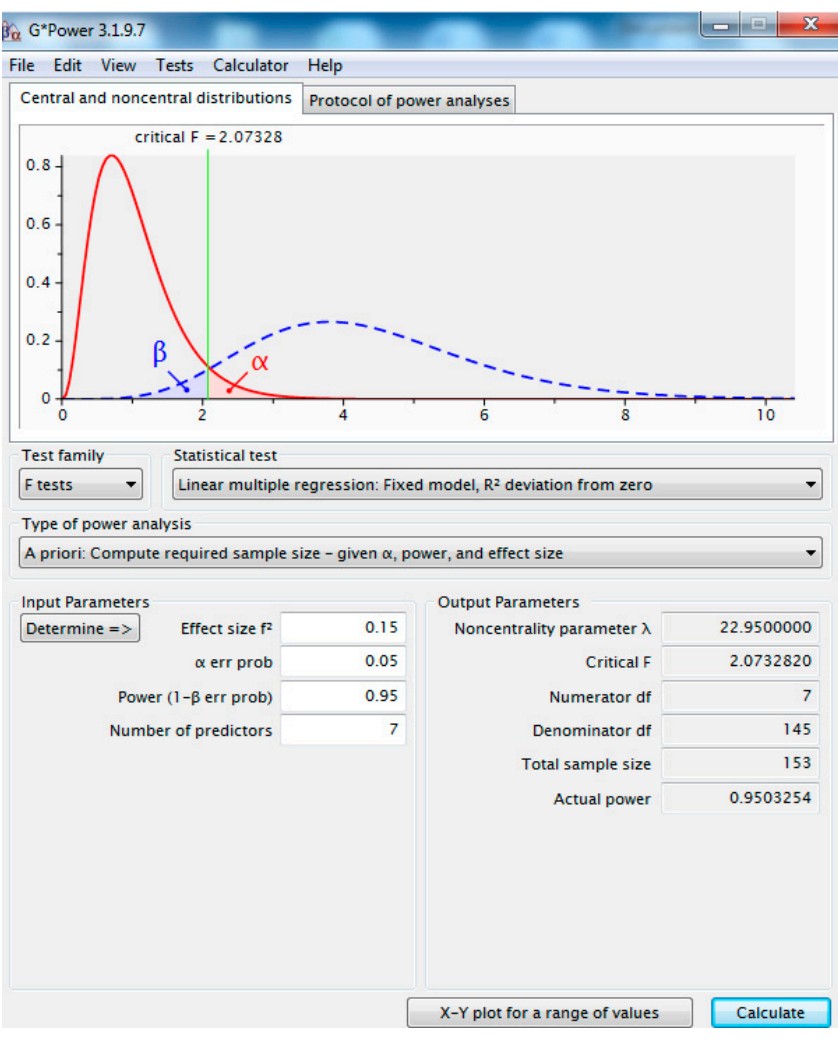

**Figure 2.** Sample size calculation using G Power 3.1.9.7.

*3.2. Measure*

The study used a quantitative design in which data were acquired from primary sources of information. A closed-ended survey questionnaire based on the 7-point Likert scale was used as the instrument. The instrument contains eight variables quantified using a seven-point Likert scale, ranging from 1—strongly disagree to 7—strongly agree. The scale for perceived severity and perceived vulnerability were adapted from Burns et al. [11]. Safa et al. [5] adapted the scale for cybersecurity awareness. Next, perceived barrier items were adapted from Li et al. [22]. Both response self-efficacy and security response efficiency were adapted from Hina et al. [64]. Meanwhile, protection habit items were adapted

from Dutton et al. [65]. Lastly, the cybersecurity protection behaviors were adapted from Anwar et al. [66].

## 4. Result and Analysis

The partial least-squares structural equation modeling (PLS-SEM) was performed using SmartPLS version 3.2.8 [67] to examine the relationship between the variables. PLS-SEM is preferable to covariance-based SEM (CB-SEM) for measurement estimation models and multigroup analysis [68]. The primary reason we employed PLS-SEM was due to the study's exploratory nature. The PLS-SEM approach was also employed in light of the current study's complex model [69], employing a predictive study [70] and data abnormalities [71]. Hence, PLS-SEMm better fit the study's aims than CB-SEM.

### 4.1. Demographic Information

The demographic data are presented in Table 1. According to the quota sampling, the majority of the respondents were middle management staff (49%), supporting staff (34%), and top management (17%). This final sampling occurred by the ratio sampling strategy.

**Table 1.** Demographic Information.

| Categories | Type | Frequency (n) | Percentage (%) |
|---|---|---|---|
| Gender | Male | 276 | 59 |
| | Female | 191 | 41 |
| Age | 21–25 | 10 | 2 |
| | 26–30 | 31 | 7 |
| | 31–35 | 64 | 14 |
| | 36–40 | 123 | 26 |
| | 41–45 | 122 | 26 |
| | 46–50 | 63 | 13 |
| | 51–55 | 38 | 8 |
| | More than 56 | 16 | 3 |
| Ethnicity | Malay | 416 | 89 |
| | Chinese | 8 | 2 |
| | Indian | 8 | 2 |
| | Sabah native | 18 | 4 |
| | Sarawak native | 14 | 3 |
| | Others | 3 | 1 |
| Position | Top Management | 78 | 17 |
| | Middle Management | 229 | 49 |
| | Support Staff | 160 | 34 |
| Type of computer protection that use at work | Antivirus | 415 | 89 |
| | Antimalware | 219 | 47 |
| | Firewall | 320 | 69 |
| | Virtual Private Network (VPN) | 220 | 47 |
| | Pop-up blockers | 229 | 49 |
| | Others | 43 | 9 |
| | I'm not using any computer protection at work | 7 | 1 |

In terms of gender, an equal distribution of males (59%) and females (41%) was achieved. Furthermore, the majority of respondents were Malay (89%), with the remainder comprising Sabah natives (4%), Sarawak natives (3%), Chinese and Indian (2% each), and others (1%). Based on the type of computer protection at work, antivirus was the most used (89%), followed by firewalls (69%), pop-up blockers (49%), antimalware and VPNs, (each obtaining 47%), and others (9%).

### 4.2. Common Method Bias

This study used a statistical technique to address a prevalent method bias that frequently occurs in behavioral research [72]. According to Kock [73], a comprehensive collinearity test can be performed to analyze common method bias in PLS-SEM. A variance inflation factor (VIF) of less than 3.3 does not exhibit common method bias. This model has no serious issues, as all latent constructs have VIF values of less than 3.3, as determined by the full collinearity test shown in Table 2.

**Table 2.** Full Collinearity.

| Construct | CSB | PB | PS | PV | PH | RSE | SRE | TA |
|---|---|---|---|---|---|---|---|---|
| VIF | 2.564 | 1.263 | 1.532 | 1.661 | 3.304 | 2.094 | 1.326 | 1.665 |

CSB = cybersecurity behavior; PB = protection barrier; PS = perceived severity; PV = perceived vulnerability; PH = protection habit; RSE = response self-efficacy; SRE = security response efficacy; TA = threat awareness.

### 4.3. Measurement Model

We tested the model built using a 2-step approach according to Anderson and Gerbing's [74] recommendations. We tested the measurement model first to ensure the validity and reliability of the instruments employed [75,76].

We evaluated the measurement model's loadings, average variance extracted (AVE), and composite reliability (CR). The loadings values should be ≥0.5, the CR should be ≥0.7, and the AVE should be ≥0.5 [75]. According to Urbach and Ahlemann [69], an indicator is dependable if it accurately measures what it is designed to measure. The dependability of indicators is determined by determining the extent to which they are consistent with the metric they are intended to measure. Typically, convergent validity can be assessed by examining the average variance extracted (AVE). Convergent validity establishes a relationship between two measures measuring the same construct. Henseler et al. [77] emphasized the importance of a construct's AVE being at least 0.5 to achieve convergent validity. Based on Table 3, four loadings on cybersecurity behavior items were less than 0.708, which was likewise acceptable [75]. The AVEs were all greater than 0.5, indicating convergent validity. The CR were all greater than 0.7, indicating acceptable internal consistency reliability.

**Table 3.** Reliability and validity analysis.

| Construct | Items | Loadings | Composite Reliability | Average Variance Extracted (AVE) |
|---|---|---|---|---|
| Threat Awareness | ACS1 | 0.843 | 0.879 | 0.644 |
| | ACS2 | 0.744 | | |
| | ACS3 | 0.803 | | |
| | ACS4 | 0.819 | | |
| Cybersecurity behaviors | CSPB1 | 0.662 | 0.891 | 0.507 |
| | CSPB2 | 0.719 | | |
| | CSPB3 | 0.785 | | |
| | CSPB4 | 0.697 | | |
| | CSPB5 | 0.645 | | |
| | CSPB6 | 0.689 | | |
| | CSPB7 | 0.768 | | |
| | CSPB8 | 0.719 | | |
| Perceived Barriers | PB1 | 0.798 | 0.904 | 0.703 |
| | PB2 | 0.901 | | |
| | PB3 | 0.873 | | |
| | PB4 | 0.774 | | |

**Table 3.** *Cont.*

| Construct | Items | Loadings | Composite Reliability | Average Variance Extracted (AVE) |
|---|---|---|---|---|
| Protection habit | PH1 | 0.889 | 0.922 | 0.575 |
| | PH2 | 0.888 | | |
| | PH3 | 0.861 | | |
| | PH4 | 0.683 | | |
| | PH5 | 0.865 | | |
| | PH6 | 0.6 | | |
| | PH7 | 0.737 | | |
| | PH8 | 0.717 | | |
| | PH9 | 0.471 | | |
| Perceived Severity | PS1 | 0.907 | 0.923 | 0.749 |
| | PS2 | 0.906 | | |
| | PS3 | 0.802 | | |
| | PS4 | 0.842 | | |
| Perceived Vulnerability | PV1 | 0.864 | 0.95 | 0.827 |
| | PV2 | 0.917 | | |
| | PV3 | 0.92 | | |
| | PV4 | 0.936 | | |
| Response Self-Efficiency | RSE1 | 0.851 | 0.892 | 0.675 |
| | RSE2 | 0.859 | | |
| | RSE3 | 0.746 | | |
| | RSE4 | 0.825 | | |
| Security Response Efficiency | SRE1 | 0.942 | 0.966 | 0.876 |
| | SRE2 | 0.954 | | |
| | SRE3 | 0.949 | | |
| | SRE4 | 0.897 | | |

The discriminant validity was then examined in step two using the HTMT criterion [77]. The HTMT values should be ≤0.85 for the stricter criterion, and the more lenient criterion should be ≤0.90 [76]. As this study is an exploratory study, all the values must be less than the HTMT0.90 threshold value, indicating good discriminant validity. Table 4 indicated that the HTMT values were lower than the stricter criterion of ≤0.85, implying that the respondents recognized the eight constructs as being separate constructs, thus indicating substantial discriminant validity.

**Table 4.** Heterotrait-Monotrait Ratio of Correlations (HTMT).

| Construct | CSB | PB | PS | PV | PH | RSE | SRE | TA |
|---|---|---|---|---|---|---|---|---|
| CSB | | | | | | | | |
| PB | 0.297 | | | | | | | |
| PS | 0.155 | 0.282 | | | | | | |
| PV | 0.076 | 0.405 | 0.624 | | | | | |
| PH | 0.888 | 0.261 | 0.141 | 0.046 | | | | |
| RSE | 0.666 | 0.171 | 0.105 | 0.08 | 0.779 | | | |
| SRE | 0.398 | 0.132 | 0.077 | 0.171 | 0.416 | 0.478 | | |
| TA | 0.601 | 0.122 | 0.204 | 0.117 | 0.667 | 0.615 | 0.399 | |

### 4.4. Structural Model Assessment

After establishing the measurement model, we proceeded with evaluating the structural model as shown in Figure 3. Bootstrapping was utilized to create results for each path relationship in the model by reinstating the initial sample to generate a bootstrap sample and provide standard errors for each hypothesis tested [70]. Chin [78] recommended performing bootstrapping with 1000 resamples concerning the number of resamples.

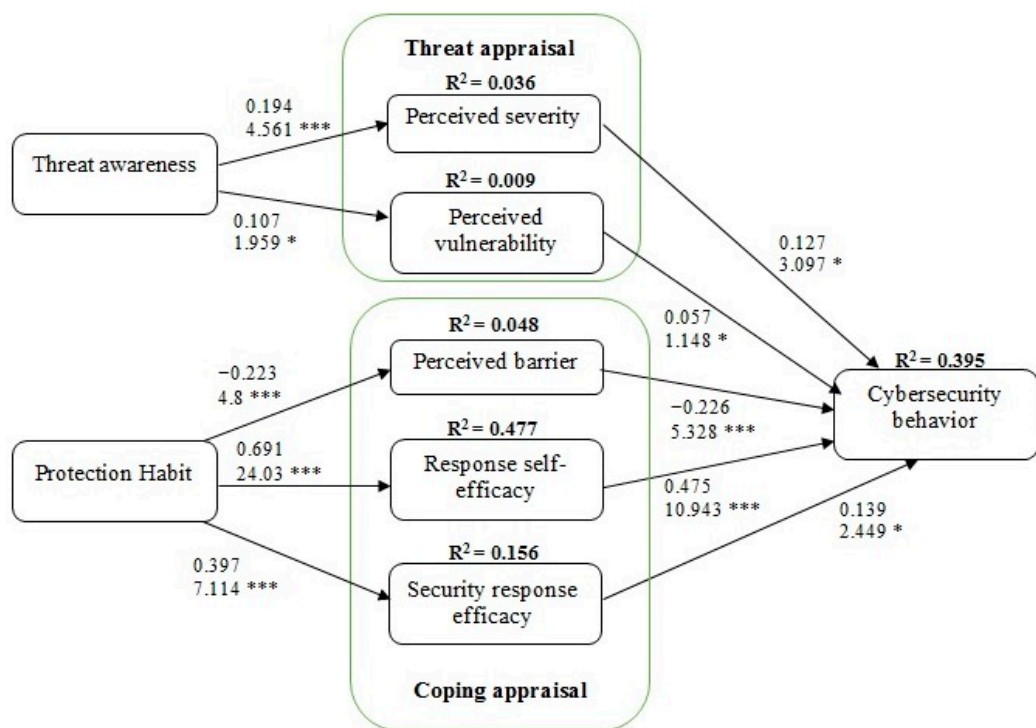

**Figure 3.** Results of the structural model. Note: * $p < 0.05$; *** $p < 0.001$.

To validate the model against the hypotheses, the path coefficient between exogenous and endogenous variables (β-value), t-values, and squared multiple correlations (R2) were analyzed to determine the explained variation on the endogenous variable. First, we tested the effects of TA on PS and PV. The R2 was 0.036 and 0.009, which showed 36% and 9% of the explained variance in CSB. PS (β = 0.194, $p < 0.01$) and PV (β = 0.107, $p < 0.01$), were positively related to CSB; thus, H1 was supported while H2 was not. Next, we tested the effects of PH on PB, RSE, and SRE. The R2 values were 0.048, 0.477, and 0.156, which showed that the explained variances of PH were 48%, 47.7%, and 15.6%, respectively. PB (β = −0.223, $p < 0.01$), RSE (β = 0.691, $p < 0.01$), and SRE (β = 0.397, $p < 0.01$), were all positively significant, hence supporting H3, H4, and H5. Lastly, the R2 for the effect of the five predictors on CSB was 0.398, indicating that the five predictors explained 39.8% of the variance in CSB. PS (β = 0.127, $p < 0.01$), PV (β = 0.057, $p < 0.01$), PB (β = −0.226, $p < 0.01$), RSE (β = 0.475, $p < 0.01$), and SRE (β = 0.139, $p < 0.01$), were all positively related to CSB, except for PV. Thus, H6, H8, H9, and H10 were supported, except for H7. Table 5 summarizes the result of hypothesis testing.

**Table 5.** Summary of the structural model.

| Hypothesis | Relationship | Path Coefficient, β | T Value | *p* Value | BCI LL | BCI UL | Effect Size, f2 | Decision |
|---|---|---|---|---|---|---|---|---|
| H1 | TA → PS | 0.194 | 4.561 | 0 | 0.128 | 0.285 | 0.039 | Supported |
| H2 | TA → PV | 0.107 | 1.959 | 0.05 | −0.007 | 0.214 | 0.012 | Not Supported |
| H3 | PH → PB | −0.223 | 4.8 | 0 | −0.318 | −0.137 | 0.052 | Supported |
| H4 | PH → RSE | 0.691 | 24.03 | 0 | 0.636 | 0.747 | 0.915 | Supported |
| H5 | PH → SRE | 0.397 | 7.114 | 0 | 0.294 | 0.507 | 0.187 | Supported |
| H6 | PS → CSB | 0.127 | 3.097 | 0.002 | 0.05 | 0.206 | 0.018 | Supported |
| H7 | PV → CSB | 0.057 | 1.148 | 0.251 | −0.049 | 0.15 | 0.003 | Not Supported |
| H8 | PB → CSB | −0.226 | 5.328 | 0 | −0.309 | −0.145 | 0.073 | Supported |
| H9 | RSE → CSB | 0.475 | 10.943 | 0 | 0.386 | 0.557 | 0.3 | Supported |
| H10 | SRE → CSB | 0.139 | 2.449 | 0.015 | 0.034 | 0.25 | 0.026 | Supported |

BCI LL = Bias confidence interval lower limit; BCI UL = Bias confidence interval upper limit.

## 5. Discussion

The PMT model incorporates two domains of coping and threat appraisals. The findings depicted that all PMT variables significantly impacted cybersecurity behavior except for perceived vulnerability. However, PMT was highlighted as one of the most convincing justifications for a person's decision to take precautions [26]. Furthermore, this study found that coping and threat appraisal positively affects the cybersecurity behavior of government employees in Malaysia. According to PMT, this is consistent with previous research indicating that individual variables can induce protective behaviors [79]. In light of this, a person's perceived vulnerability is an assessment of whether they believe they could be at risk for dangers [32]. Moreover, Ifinedo [3] discovered that perceived severity is utilized to gauge how well users comprehend the severe repercussions of potentially harmful cybersecurity acts. Meanwhile, consumer confidence in the danger of cyberthreats is gauged using perceived vulnerabilities. Employees' perceptions of the severity and vulnerability of cybersecurity incidents are also positively impacted [22].

Regardless of technological advancements, the human aspect is considered the security system's weakest link due to people's ignorance and lack of security concerns [80]. This study discovered that threat awareness positively influences users' perceived severity. Similar to the recent study by Li et al. [22], perceived severity is a judgment made when confronted with a threat on cyber-related activities. In order to mitigate security risks, security awareness courses and training are required. Meanwhile, this study discovered that threat awareness does not influence a user's perceived vulnerability. According to Aldossary and Zeki [81], internet users with moderate security awareness use weak passwords, open email attachments from unfamiliar senders, and other unprotective behavior. Even though they were aware of the risks, they undervalued them.

The most significant factors to consider when analyzing behavioral change are the perceived barriers [40]. They contribute to individuals' perceptions of the cost and difficulty of conducting cybersecurity activities [25]. Individuals' perceptions of cybersecurity data protection barriers are based on their prior experiences, and it was observed that the low barriers they perceived were influenced by their previous experiences. Ye and Potter [82] proved that personal habits could mitigate the impact of other beliefs on certain services. Meanwhile, Barnes and Boheinger [83] discovered that habit is a significant predictor of continuing use intention for internet users. Thus, this study discovered that protection habits negatively influence users' perceived barriers.

A habit may be seen as an automatic behavioral response generated by a situational stimulus that does not require cognitive processing due to the learned relationship between one behavior and pleasing outcomes [84]. As a result, habit formation necessitates an inevitable repetition or practice level [85]. Once a habit is established, behavior becomes automatic [86]. However, the extent to which habits are utilized in study models differs. While some researchers argue that habits act as moderators in the link between intention and its determinants [87,88], others assert that habits have a direct effect on the intention to use the internet [83,89].

Additionally, Vance et al. [1] demonstrate that reaction efficacy is the belief in the perceived benefits of the coping behavior due to the danger being removed. Meanwhile, self-efficacy refers to an individual's belief in his or her ability to carry out protective behaviors. As a result, this study discovered that regular compliance with cybersecurity policies benefits self-efficacy and response efficacy.

This study finds that perceived barriers have a negative influence on online security behaviors as expected. The construct reflects an individual's fear of the challenges of adopting a new behavior, and it is directly related to proactive security behaviors [90]. It was also posited that if habits could be deactivated as quickly as they are developed, they would not provide a barrier to changing behavior [91]. Once developed, a habit affects how we look for and process information. Finally, user resistance results from perceived barriers such as inconvenience, time commitment, and habit change.

## 6. Implications

### 6.1. Theoretical Implications

This study emphasizes the protective coping appraisal of employees' cybersecurity behavior. The study reported similar findings on the reliability of PMT in understanding and predicting employee security behavior as reported in the literature [1,3,21,27,28,30,44]. Meanwhile, Burns et al. [11] claimed that cybersecurity behavior has largely overlooked positive coping mechanisms, such as self-efficacy, and has framed security motivation exclusively in terms of fear appeals. We discovered that the coping features of PMT were more significant on an employee's cybersecurity behavior than threat appeals. Hanus and Wu [28] observed that three coping appraisal variables (perceived barrier, self-efficacy, and response-efficacy) were significant predictors of reported security behavior.

In contrast, the threat elements (perceived severity and perceived vulnerability) did not. As a result, government employees in Malaysia can engage in specific behaviors and assess their capacity to implement the cybersecurity behaviors outlined. However, they are ignorant of the threat to user perceptions since they do not consider themselves vulnerable to cyberthreats.

A habit is a significant predictor of human behavior and conduct. Cybersecurity behavior research can help provide new viewpoints to render more comprehensive explanations of the mechanisms behind establishing cybersecurity habits [48]. Furthermore, situational conditions play a significant role in habit formation, and the frequency with which a behavior is displayed cannot entirely explain the self-consciousness that habits imply [39]. Thus, in this study, we attempted to define habit formation in terms of the influence of significant situational factors, which refer to the range of relevant situations that an individual has previously faced.

This study contributes significantly to the body of knowledge by presenting the theory of cybersecurity behavior as a strong predictor of coping and threat evaluation processes. This demonstrates that it is not sufficient to establish a "fear appeal" [45] in the hope of inspiring people to act when undertaking threat awareness. Users are more likely to be motivated if they are aware of and confident in using precautionary measures. While technology has increased both locally and globally in recent years, effective cybercrime awareness must be considered [92]. This study concludes that Malaysian government employees are accountable for increasing cybersecurity awareness among their constituents. As a result, governments must contribute to the fight against cybercrime through their different authorities and organizations.

This section may be divided by subheadings. It should provide a concise and precise description of the experimental results, their interpretation, as well as the experimental conclusions that can be drawn.

### 6.2. Practical Implications

The outcome of this study would provide empirical insights into reducing the unfavorable consequences of cyberthreats on the user in Malaysia. Policymakers and government bodies such as the Malaysian Communications and Multimedia Commission can create effective regulations on human behavior relating to direct connection with cyberconnected practices. On the ground, individuals within organizations who interact closely with end users (network engineers, programmers, and others) would determine the most critical antecedents with high risk or associated aspects that can withstand cyberattacks.

This study's findings may help protect society from cyberattacks such as phishing, fraud, and other cyberrisks. As the region's most susceptible country, Malaysia needs a comprehensive solution to this issue to prevent being readily targeted by criminals and fraudsters. Aside from technological and physical safeguards, governments and industry partners should intervene and establish methods to increase people's cybersecurity protection. Furthermore, investors want to invest in countries that are perceived to be safe and stable. The country's goal of being recognized as a haven for cyberusers may be accomplished by increasing cybersecurity via user behavior.

## 7. Recommendation for Future Works

Even though the level of technology has gone up in the last few years worldwide, the awareness of cybercrime needs to be effectively taken into account. This study examines the factors influencing Malaysia's employees' cybersecurity behavior using PMT. It is used to elucidate the individual beliefs that contribute to the adoption of security behavior and how instruction influences them. The proposed model is a good predictor of security behavior. In this sense, our research lays the groundwork for fostering acceptable long-term individual behavior in system security.

Besides that, future work should dig deeper into the knowledge gap regarding cybersecurity behaviors to adopt and evaluate the efficiency of various coping messages for individuals with varying levels of cybersecurity literacy. This would necessitate a more nuanced application of PMT to comprehend how interventions might target self- and response-efficacy knowledge and beliefs. This study also provides insight into how small behavioral encouragement can result in more significant security effects.

## 8. Conclusions

From the perspectives of PMT, threat awareness, and habit, this study shed insight into how government personnel behave in the face of cyberattacks. The rising usage of tablets and smartphones for personal and financial information underscores the need for more information security research focusing on cybersecurity behavior. The study adequately predicts Malaysian government employee cybersecurity behavior. Factors ranging from threat knowledge to perceived severity and vulnerability, protection habits to the perceived barrier, self-efficacy, and response efficacy contribute to a government employee of Malaysia's cybersecurity behavior. Furthermore, cybersecurity is a crucial issue in today's organizations, especially in a government department where in certain departments there are sensitive and highly confidential data related to national security. Cybersecurity measures, particularly human behavior, must be further reinforced to ensure the Malaysian government is regarded as a cybercrime-protected government agency.

**Author Contributions:** Conceptualization, N.S.S. and M.A.F.; methodology, N.S.S. and M.A.F.; software, N.S.S. and M.A.F.; validation, N.S.S. and M.A.F.; formal analysis, N.S.S. and M.A.F.; investigation, N.S.S. and M.A.F.; resources, N.S.S. and M.A.F.; data curation, N.S.S.; writing—original draft preparation, N.S.S.; writing—review and editing, M.A.F., W.W. and S.H.; visualization, N.S.S. and M.A.F.; supervision, M.A.F.; project administration, N.S.S.; funding acquisition, M.A.F. All authors have read and agreed to the published version of the manuscript.

**Funding:** Ministry of Higher Education Malaysia under the Fundamental Research Grant Scheme FRGS RACER/1/2019/SS03/UMP//1 (University Grant no. RDU192619).

**Institutional Review Board Statement:** Not applicable.

**Informed Consent Statement:** Not applicable.

**Data Availability Statement:** Not applicable.

**Conflicts of Interest:** The authors declare no conflict of interest.

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
