# Peer review of "Cybersecurity Behavior among Government Employees: The Role of Protection Motivation Theory and Responsibility in Mitigating Cyberattacks"

_information, doi:10.3390/info13090413_

Round 1

Reviewer 1 Report

Page 1- Line 26 to 32

The manuscript contains some Informational instructions that need to be removed.

Page 3- Line 147

Move heading “ 2.2.2 Threat awareness and perceived vulnerability” down to the next page.

Page 4 – Line 164

“Integrating habits within the coping appraisal domain has been shown 163 to predict individual cyber security behaviour (Verplanken & Orbell, 2003). …………………… Perceived”

There is a huge place gap before the word “perceived” that needs to be closed.

Page 5 – Line 250

Move heading “2.2.9 Response self-efficacy and cyber security behaviour “down to the next page.

Page 6

“Figure 1. Research model” is not fully displayed. Figure 1 needs to be reuploaded so it can show adequately.

Page 7- Methodology

The authors expeditiously described the sampling method. The authors must thoroughly indicate the population size and explain how the sample size was calculated.

Page 11

Table 5 needs to be expanded to show its entire content.

Page 12

“Figure 2. Results of the structural model” are not fully displayed. Figure 2 needs to be reuploaded so it can show adequately.

Reviewer 2 Report

1.       In the review of the literature, it is more clearly justified why it is recommended to apply the PMT model.

2.       Modify graphs 1 and 2 are not complete

3.       Explain with a mathematical formula how I defined the population to take the sample to collect the information to carry out the survey of the 446 people.

4.       Determine why the PMT model does not affect vulnerabilities

5.       Check that the abstract and all the phases of the article comply with the structure of the IMRYD methodology

6.       If possible, remove or replace all references older than 5 years to ensure the reliability of the information used
